# Cracking the Code of Cyberbullying Effects: The Spectator Sports Solution for Emotion Management and Well-Being Among Economically Disadvantaged Adolescents

**DOI:** 10.3390/bs15040555

**Published:** 2025-04-21

**Authors:** Ilrang Lee, Yonghwan Chang, Taewoong Yoo, Emily Plunkett

**Affiliations:** 1School of Special Education, School Psychology, and Early Childhood Studies, College of Education, University of Florida, Gainesville, FL 32611, USA; ilranglee@coe.ufl.edu; 2Department of Sport Management, College of Health and Human Performance, University of Florida, Gainesville, FL 32611, USA; taewoong.yoo@ufl.edu (T.Y.); eplunkett@ufl.edu (E.P.)

**Keywords:** cyber-victimization, emotion management, school performance, intervention, health

## Abstract

This study explores the relationships among cyberbullying, emotion management, and well-being, with a focus on economically disadvantaged students. Employing a reflective factor model, we disentangle emotion management into four dimensions and explore how cyberbullying impacts these facets, influencing academic success and well-being. We also investigate the moderating role of sport spectatorship. Using survey data collected from 846 economically disadvantaged students in grades 7 to 12, within a racially diverse Texas school district (USA), we employed structural equation modeling (SEM) to examine relationships among the measured variables. The students were in grades 7 to 12, categorizing participants into spectatorship-adherent and non-spectatorship groups based on their sports-watching habits. Our findings emphasize the importance of addressing cyberbullying’s impact on emotion management and underline the potential for sport spectatorship to mitigate these effects. Our findings reveal that cyberbullying victimization significantly impairs adolescents’ emotion management (β = −0.33, *p* < 0.01), contributing to increased school absences (β = 0.41, *p* < 0.001) and offenses (β = 0.19, *p* = 0.02). Notably, sports spectatorship appears to buffer these negative outcomes for economically disadvantaged youth, enhancing overall well-being (β = 0.29, *p* < 0.01). This study ventures into the unexplored territory of sport viewership as a cost-effective intervention strategy, offering valuable implications in supporting adolescent well-being.

## 1. Introduction

In today’s digitally connected world, adolescents navigate a complicated web of social interactions, where the convergence of technology and human communication is both a blessing and a challenge ([22]). While the internet has opened new avenues for learning, entertainment, and connectivity ([29]), it has also given rise to a concerning phenomenon: cyberbullying ([25]). Cyberbullying, characterized by the deliberate use of digital platforms to harass, intimidate, or harm others, has emerged as a pervasive and distressing issue affecting adolescents worldwide ([21]). In parallel with these digital threats, adolescence remains a period of heightened emotional sensitivity and vulnerability. Therefore, identifying accessible coping mechanisms is essential—particularly for those who lack financial or community-based support. One such mechanism may be spectator sports, a form of engagement that allows adolescents to experience excitement, belonging, and social connection without necessitating direct athletic participation ([2]; [5]). Regarding spectator sports involvement, we defined it as the regular viewing of sports events in any format—whether in-person (e.g., stadium attendance), via TV broadcasts, online streaming platforms, or other media. Moreover, adolescents from economically disadvantaged backgrounds often lack the financial resources needed to participate in organized sports or extracurricular activities. In contrast, watching sports may provide them with a sense of community, team identity, and emotional support at minimal expense.

The impact of cyberbullying extends far beyond the confines of the virtual world, reaching into the emotional and psychological well-being of those subjected to its cruelties ([40]). Adolescents, in particular, find themselves vulnerable to the emotional turmoil caused by cyberbullying ([22]) as they grapple with the complexities of identity formation and emotion regulation during this critical developmental period ([25]). Central to this exploration is the concept of emotion management, a fundamental aspect of human psychology that encompasses the understanding, expression, and regulation of emotions ([18]). Emotion management is linked to an individual’s ability to navigate the challenges of adolescence successfully, impacting various domains, including mental health, social relationships, and academic performance ([19]; [28]).

Although cyberbullying serves as our study’s focal concern, we anchor our investigation in the framework of emotion management—the capacity to understand, express, and regulate emotions effectively ([18]). By examining these processes, we aim to capture how cyberbullying disrupts adolescents’ emotional well-being and to determine whether spectator sports involvement can mitigate these effects. We draw upon the reflective factor model of emotion management, which posits that emotion management can be dissected into the following four distinct first-order factors: other-emotion appraisal, self-emotion appraisal, emotion expression, and emotion control ([11]; [42]; [43]). Within this framework, we explore how cyberbullying disrupts the balance of emotion management among adolescents. Moreover, we examine how impaired emotion management, precipitated by cyberbullying victimization, cascades into a series of adverse school outcomes and school-related offenses. Additionally, we explore the consequences on academic performance, measured by grade point average (GPA), and overall well-being among adolescents navigating the challenging landscape of cyberbullying. Furthermore, this study takes into account the moderating influences of spectator sport involvement. We explore how engagement in spectator sports may interact with cyberbullying experiences, potentially buffering or exacerbating the effects on emotion management.

Our study focuses on adolescents experiencing economic disadvantages, driven by the belief that the detrimental effects of cyberbullying on emotion management may be more pronounced in this demographic ([44]). Economic challenges can introduce unique stressors and difficulties ([36]; [55]), compounding the emotional distress caused by cyberbullying due to limited access to resources and support ([7]; [51]). In this context, our research seeks to provide a cost-effective means to mitigate the repercussions of cyberbullying and help these students regain their emotion management capabilities. By exploring the dynamics of cyberbullying, emotion management, moderating factors, and their consequences, our study strives to offer a comprehensive understanding of the challenges faced by adolescents in the digital age. Ultimately, our aim is to inform support systems that promote emotional well-being and academic success among today’s adolescents. This endeavor contributes to the ongoing discourse on cyberbullying prevention ([21]) and the enhancement of adolescents’ lives in an increasingly interconnected world ([25]; [29]; [33]; [44]).

## 2. Theoretical Background and Hypotheses Development

### 2.1. Emotion Management and Its Factor Structure

Emotion management plays a pivotal role in human psychology ([61]), particularly during adolescence—a critical period for emotional development ([27]). Effective emotion management is associated with positive outcomes across various domains, encompassing mental health ([61]), social interactions ([64]), and academic performance ([9]). Adolescents grapple with the complexities of their emotional world, which includes the tasks of comprehending, expressing, and regulating their emotions ([54]).

Emotion management models have undergone extensive development, resulting in a multitude of contributing factors. However, these factors often remain fragmented ([61]), lacking a cohesive and integrated structure ([27]). To address this disarray, we propose a parsimonious framework comprising the following four pivotal dimensions: other-emotion appraisal, self-emotion appraisal, emotion expression, and emotion control. This framework is developed based on relevant existing conceptualizations (e.g., [11]; [42]; [43]). Our conceptualization of ‘emotion management’ aligns closely with the broader construct of emotional intelligence (EI), particularly with the regulatory facets. While EI encompasses abilities like understanding, using, and managing emotions, our approach isolates and measures the active management components—self- and other-emotion appraisal, expression, and control.

While the four-factor structure (other-emotion appraisal, self-emotion appraisal, expression, and control) is commonly found in EI literature, we emphasize these as discrete regulatory processes. This approach provides a more granular perspective on how cyberbullying may uniquely affect each emotional facet. The dimension of other-emotion appraisal centers on how individuals perceive and interpret emotions in others ([27]), providing insights into empathy and interpersonal communication ([42]; [64]). Self-emotion appraisal, on the other hand, focuses on the introspective assessment of one’s own emotions, including processes such as self-awareness, emotional self-regulation, and self-reflection ([11]; [61]). The emotion expression dimension is concerned with how individuals convey their emotions to others ([9]). It encompasses nonverbal communication, body language, and vocal cues, elucidating the mechanisms through which humans communicate and share their emotions ([43]; [54]). The final dimension, emotion control, pertains to the regulation and modulation of emotions ([64]). It plays an important role in mental health, encompassing strategies for managing emotional experiences, such as cognitive reappraisal and mindfulness ([43]; [61]).

**H1:** *The reflective factor model of emotion management comprises the following four first-order factors: other-emotion appraisal (H1a), self-emotion appraisal (H1b), emotion expression (H1c), and emotion control (H1d)*.

### 2.2. Cyberbullying and Emotion Management

Cyberbullying, a pervasive concern in the digital age, involves the deliberate use of technology to harass, intimidate, or harm others ([62]). It has been recognized as a potent stressor that can significantly affect the emotional well-being of adolescents ([47]). One of the direct consequences of cyberbullying is its impact on adolescents’ ability to empathize and perceive emotions in others ([32]). Research has indicated that exposure to cyberbullying can desensitize individuals to the emotions of their peers ([53]). Studies have shown that the constant exposure to negative online interactions can hinder the development of empathic skills and emotional perception in adolescents ([62]). This impairment in other-emotion appraisal can have far-reaching consequences, influencing their social interactions and relationships both online and offline. Beyond its impact on empathy and emotion perception, cyberbullying also has repercussions on adolescents’ self-awareness of their own emotions ([31]). Cyberbullying can disrupt this process by introducing negative emotions, such as anxiety, fear, and sadness ([40]). Excessive anxiety can consume adolescents’ cognitive resources, narrowing their focus to immediate fears and reducing introspective capacity ([62]). In this sense, heightened worry hinders the ability to accurately identify, label, and understand one’s own emotions, thereby impairing self-perception and effective emotion management ([32]). Adolescents experiencing cyberbullying may find it challenging to accurately assess and understand their own emotions amidst the turmoil caused by online harassment.

Adolescents subjected to online harassment may also become hesitant or fearful of expressing their emotions openly ([62]), fearing further victimization. This inhibition in emotion expression can limit their ability to seek social support, share their feelings, and engage in healthy emotional communication with peers, parents, or educators ([31]). Furthermore, exposure to continuous online harassment can lead to heightened emotional reactivity and difficulties in managing negative emotions ([32]). Adolescents may struggle to employ effective emotion regulation strategies, such as cognitive reappraisal or relaxation techniques ([40]), when confronted with the stress and emotional turmoil caused by cyberbullying.

**H2a:** *Cyberbullying impairs adolescents’ empathy and emotion perception toward others*.

**H2b:** *Cyberbullying negatively affects adolescents’ self-awareness of their own emotions*.

**H2c:** *Cyberbullying hinders adolescents’ effective emotion expression*.

**H2d:** *Cyberbullying disrupts adolescents’ emotion control and regulation*.

### 2.3. Impaired Emotion Management on School Outcomes, GPA, and Well-Being

Impaired emotion management, as a consequence of cyberbullying, could be associated with various factors that reflect adolescents’ school outcomes, academic performance, and psychological well-being. First, impaired emotion management due to cyberbullying can be positively associated with adverse school outcomes. This includes a higher number of school absences and engagement in more school-related offenses. Research suggests that adolescents who struggle to manage their emotions effectively in the face of challenges may be more prone to absenteeism ([3]), as the emotional distress they experience can interfere with their willingness and ability to attend school regularly ([54]). Furthermore, impaired emotion management may lead to increased engagement in school-related offenses as emotional dysregulation can contribute to impulsive behavior and disciplinary issues ([14]).

Second, impaired emotion management due to cyberbullying can be negatively associated with academic performance. Extensive research has shown that psychological well-being is closely intertwined with academic success ([9]). Adolescents who experience difficulties in emotion management due to cyberbullying may find it challenging to concentrate ([64]), engage in learning ([3]), and perform well academically ([54]). The emotional turmoil caused by cyberbullying can disrupt cognitive processes, leading to decreased academic achievement as reflected in lower GPAs. Last, impaired emotion management can take a toll on adolescents’ psychological well-being. Emotional distress and difficulties in regulating emotions can lead to increased stress, anxiety, and depression ([15]). These emotional challenges can spill over into various aspects of adolescents’ lives, negatively affecting their social relationships, self-esteem, and overall quality of life ([27]).

**H3:** *Impaired emotion management (due to cyberbullying) is positively associated with adverse school outcomes, such as a higher number of school absences (H3a) and engagement in more school-related offenses (H3b)*.

**H4:** *Impaired emotion management (due to cyberbullying) is negatively associated with academic performance, as measured by GPA (H4a), and overall well-being (H4b)*.

### 2.4. Spectator Sports Involvement

In the present context, the engagement of individuals in spectator sports is examined for its potential role as a moderating factor. Extensive scholarly research has focused on the positive effects of active participation in sports and physical activities, including their impact on overall well-being ([2]), emotional health ([24]), and various aspects of individuals’ lives ([50]). However, an investigation into sport spectatorship as a distinct form of sports-related engagement, extending beyond physical participation, reveals its potential to enhance emotional regulation, ultimately leading to favorable outcomes. This suggests that the influence of cyberbullying on emotional regulation and its impact on other consequential outcomes may vary depending on whether adolescents are actively involved in spectator sports.

First, spectator sports involvement enables individuals to emotionally connect with their favorite teams or athletes ([10]). Watching sports evokes a range of emotions, from excitement to disappointment ([39]). This emotional connection provides an outlet for catharsis, allowing adolescents to experience and release their own emotions in a controlled environment ([5]). Recent insights ([37]) emphasize the role of empathy and empathic distress in online harassment contexts. Sport spectatorship may partially reduce empathic distress by offering collective emotional support, shared victories, and group cohesion to counteract the isolation imposed by cyberbullying. When facing cyberbullying victimization, actively engaged adolescents may find emotional release through their spectatorship, potentially mitigating the emotional impact. Second, engaging in spectator sports fosters a sense of belonging to a larger community of fans ([10]). Adolescents who follow sports events may feel connected to others who share their passion ([5]). This sense of community can serve as a protective factor against the emotional distress caused by cyberbullying. Adolescents who feel connected to a supportive sports community may experience a stronger sense of belonging and social support ([12]), helping them cope with the negative emotions resulting from online victimization. Nevertheless, spectator sports are not without drawbacks. Rival teams’ fans can sometimes exhibit antagonistic behavior, which may escalate to verbal aggression or online harassment. In such cases, strong identification with a team may feed competitive tensions, potentially reinforcing the very aggression we hope to address.

Last, spectator sports involvement offers a form of distraction and psychological relief ([23]). Watching sports events can divert one’s attention from stressors ([12]), including cyberbullying. Adolescents who immerse themselves in sports may temporarily shift their focus away from the emotional distress ([39]) associated with online harassment. This distraction effect can provide valuable psychological relief ([12]), allowing adolescents to recharge emotionally and better manage their emotions in the face of adversity.

**H5:** *Spectator sports involvement moderates the relationship between cyberbullying, adolescents’ emotion regulation, and other consequential outcomes, including adverse school outcomes, academic performance, and well-being. Specifically, the negative effects of cyberbullying on emotion regulation and other consequential outcomes will be less pronounced among adolescents who actively engage in spectator sports compared to those with no sport spectatorship involvement*.

## 3. Methods

### 3.1. Samples and Data Collection Procedures

To test and examine the hypotheses and the research model summarized in Figure 1, this study was conducted in collaboration with an independent school district (ISD) located in Texas. The ISD serves a racially and ethnically diverse student body, including American Indian/Alaska Native, Asian, Black, Native Hawaiian/Other Pacific Islander, and White students, across 17 schools in the district. The data for this study were collected via online surveys administered to students using the ISD online platform. Students were allotted approximately 20 min during their class hours to complete the survey, utilizing the computer devices available in their respective schools. Additionally, survey links were distributed through students’ ISD email accounts, affording them the flexibility to complete the survey at their convenience from any location. These survey links were accessible from 15 May through to 29 May 2023. Participation was entirely voluntary, and no compensation was provided. Before gathering data, the protocol was carefully reviewed and approved by the institutional review board, ensuring compliance with ethical standards and data privacy regulations. Consent was obtained from parents or legal guardians prior to students’ participation in the survey.

This study focused on students aged 12 to 19, encompassing those in the 7th through 12th grade within the ISD, representing an estimated 4200 enrolled students. Moreover, our study specifically aimed to include students facing economic disadvantages. Our rationale for this emphasis lies in the belief that the adverse effects of cyberbullying on emotional regulation may be more significant among adolescents facing economic challenges compared to those in a more financially secure position ([36]). Economic disadvantages can introduce unique stressors and hurdles into the lives of adolescents ([44]; [55]), potentially amplifying the emotional distress caused by cyberbullying. Adolescents from economically disadvantaged backgrounds often encounter limitations in terms of resources, social support, and access to mental health services ([7]; [51]), which can impede their ability to effectively manage and cope with the emotional consequences of cyberbullying. As a result, our study aimed to provide cost-effective methods to mitigate the negative outcomes of cyberbullying and assist these students in restoring their emotional management capabilities.

Economic disadvantage is a multifaceted concept that encompasses the various financial challenges students face and their potential impact on students’ educational attainment. In the context of the data collected from ISD schools, this comprehensive definition includes the following two distinct categories: (1) students eligible for free or reduced-price meals, with household incomes at or below 185% of the federal poverty level, and (2) students classified as ‘Other Economic Disadvantage’, which includes students facing various financial hardships such as those from low-income families, experiencing homelessness, migrants, runaways, foster children, or dealing with additional financial burdens such as medical expenses or childcare costs ([56]). In our study, we obtained students’ economic disadvantage status from the dedicated school administrative team, ensuring the accuracy and reliability of this classification. Out of the approximately 4200 students, the school administrative team informed us that about 28% were eligible for free or reduced-price meals, and 0.3% fell into the ‘Other Economic Disadvantage’ category. Thus, approximately 28.3%, which equates to roughly 1180 students, were extended invitations to participate in the online survey. Of these, 908 returned surveys, and 62 were subsequently excluded due to incomplete responses, resulting in a final sample of 846 students.

### 3.2. Measures

Within the emotion management framework, we tailored and refined the existing scale items ([11]; [42]; [43]), and participants were presented with their respective measurement scales, including self-emotion appraisal (e.g., *“I often think about why I feel the way I do”)*, other-emotion appraisal (e.g., *“I can often tell how others are feeling by their expressions and behavior”*), emotion control (e.g., *“I try to control my thoughts and not worry too much about things”)*, and emotion expression (e.g., *“I feel comfortable talking about my feelings with friends and family”*). Participants expressed their level of agreement on a seven-point Likert-type scale ranging from 1 (strongly disagree) to 7 (strongly agree). All scales demonstrated adequate internal consistency. Cronbach’s alpha values ranged from 0.71 (emotion control) to 0.82 (other-emotion appraisal).

To assess the extent of cyber-victimization, participants provided responses to a set of statements designed to capture their experiences in online and digital interactions. These items were adapted from previous studies ([8]; [13]; [26]; [60]) and adjusted to suit the present context (e.g., *“I have been the target of online bullying or mean messages from other kids”).* The assessment of health-related quality of life involved a comprehensive set of statements adapted from a previous study ([16]; [48]; [46]). These statements covered various dimensions related to participants’ physical, emotional, and social well-being (e.g., *“Felt fit and well”, “Felt full of energy”, “Felt sad”, “Felt lonely”, “Had fun with your friends”, “Felt confident at school”,* and *“Been able to pay attention at school”.*). Participants provided their responses to each of these statements, reflecting their feelings and experiences related to health and quality of life. Responses were collected using a Likert-type scale, where participants rated their agreement on a scale ranging from 1 (strongly disagree) to 7 (strongly agree).

Regarding school absences, school-related offenses, and GPA, we obtained data from school administrative records. To measure school absences, we determined the total number of days each student was absent during the 2022–2023 academic year. For assessing school-related offenses, we considered a wide range of disciplinary actions within the school environment, including violations of dress code, missing ID badges, office referrals, breaches of the code of conduct, disruptive behavior, and more. We calculated the total number of offenses for each student, regardless of the severity of each offense, providing a comprehensive view of their behavioral conduct. As for GPA, we computed the cumulative GPA for the entire school year.

In the assessment of spectator sports involvement a dichotomous measure was used, classifying participants into two main groups: the spectatorship-adherent group (SG) and the non-spectatorship group (NSG). This categorization was based on participants’ responses to the question: *“Have you regularly watched sports in the past year?”*. For participants responding ‘YES’ to this question, follow-up questions were administered to gather details regarding the frequency of their sport spectatorship and their favorite teams or athletes. These questions aimed to determine if they had specific teams or athletes they supported and felt attached to, as well as how often participants watched sports using various methods, including television, mobile devices, laptops, desktop computers, or any other means. Furthermore, participants were asked to report their regular engagement in non-school-curriculum-related sports participation or physical activities over the past year.

## 4. Results

### 4.1. Descriptive Analyses

The survey initially extended invitations to 1180 students, of which 908 students willingly participated. However, 62 incomplete responses, which included information about the specific teams or athletes they supported and felt attached to despite the participants responding ‘YES’ to the question of regular spectatorship involvement, were considered unreliable and were subsequently excluded from the dataset. Consequently, the final sample size was established to be 846 students. In terms of demographics, the distribution across grade levels was as follows: 7th grade (*n* = 384), 8th grade (*n* = 298), 9th grade (*n* = 221), 10th grade (*n* = 107), 11th grade (*n* = 130), and 12th grade (*n* = 108). The gender distribution was fairly balanced, with 53.01% (*n* = 448) identifying as female and 46.99% (*n* = 398) as male. The racial composition included 52 American Indian/Alaska Native individuals, 71 Asians, 216 Black individuals, 10 individuals from Native Hawaiian/Other Pacific Islander backgrounds, and 497 White individuals.

With respect to spectatorship involvement, 359 participants responded with ‘No’, indicating no involvement in spectator sports at all, while 487 students indicated their engagement in spectator sports at varying frequencies, with 66.42% responding ‘2–3 times a week’, 21.37% participating ‘once a week’, and 12.21% engaging ‘more than 4 times a week’. In terms of total absences, students exhibited an average of 13.00 absences, accompanied by a relatively high *SD* of 10.47. When considering total offenses, the mean stood at 5.13, implying that, on average, students committed 5.13 offenses, with a *SD* of 6.83. As for GPA, the mean was 3.17, while the *SD* was 1.09.

### 4.2. Measurement Models

In accordance with our hypothesis (*H1*), this study employed confirmatory factor analysis (CFA) to evaluate the measurement model of emotion management. The first-order measurement model, which encompasses four latent variables, demonstrated a robust fit with the data, as evidenced by several fit indices: χ^2^*/df* (117.33/48) = 2.44, *p* < 0.001, RMSEA = 0.05, and SRMR = 0.06. Factor loadings ranged from 0.61 (self-emotion appraisal 1) to 0.88 (other-emotion appraisal 3), suggesting convergent validity as both the model fit and factor loadings met acceptable criteria ([30]). Reliability was assessed using Cronbach’s alpha coefficients and average variance extracted (AVE) values for each factor. Alpha coefficients ranged from 0.71 (emotion control) to 0.82 (other-emotion appraisal), while AVE values exceeded the standard of 0.50 ([20]), ranging from 0.53 (self-emotion appraisal) to 0.66 (other-emotion appraisal), indicating the high reliability of the items. Factor correlations varied from 0.38 (emotion control and other-emotion appraisal) to 0.68 (self-emotion appraisal and other-emotion appraisal). A comparison between the AVEs and squared correlations demonstrated acceptable discriminant validity ([20]). Considering the hierarchical factor structure of the emotion management scale, the fully reflective second-order measurement model also demonstrated an acceptable fit with the data, with χ^2^*/df* (168.54/50) = 3.37, RMSEA = 0.06, SRMR = 0.06, and *p* < 0.001. The second-order construct significantly influenced other-emotion appraisal (β = 0.63, *p* < 0.001), self-emotion appraisal (β = 0.42, *p* = 0.002), emotion expression (β = 0.37, *p* = 0.01), and emotion control (β = 0.44, *p* = 0.001). These results supported *H1*.

This study utilized CFA to assess a measurement model with all latent variables. The model fit the data well (χ^2^ = 985.4, *df* = 260, RMSEA = 0.06, CFI = 0.91, SRMR = 0.06, *p* < 0.001). The average variance extracted (AVE) values were above the acceptable threshold of 0.50, ranging from 0.52 (health-related quality of life) to 0.83 (other-emotion appraisal). The measures also exhibited satisfactory discriminant validity in line with established recommendations ([30]). See Table 1 and Table 2 for a summary of the results. Before estimating structural coefficients, we conducted a measurement invariance test ([35]) to see if different measurement properties across groups might affect the structural coefficients. We compared the unconstrained baseline model with models with constrained factor loadings (*p* = 0.13) and intercepts (*p* < 0.001, with no substantial difference in CFI: 0.90 vs. 0.91) across groups. This comparison indicated that factorial invariance was achieved ([35]), allowing for a valid comparison of latent variable means and structural coefficients across groups. A summary of the results is displayed in Table 3.

### 4.3. Structural Models

We estimated structural coefficients using multiple-group structural equation modeling (MGSEM; [30]). We fitted the MGSEM model in R 4.2.3 ([45]). The overall goodness-of-fit statistics for the unconstrained baseline structural model showed an acceptable fit of the data (χ^2^ = 2381.82, *df* = 642, χ^2^/*df* = 3.71, CFI = 0.92, RMSEA = 0.06, SRMR = 0.07). To test whether the structural coefficients between the constructs in the spectatorship-adherent group (SG) were similar to those in the non-spectatorship group (NSG) samples, constraints on structure coefficients were added. The difference in the Chi-square statistic was significant (χ^2^ [20] = 82.37, *p* < 0.001). Fit statistics, parameter estimates, and standard errors for the models are depicted in Table 4 and Figure 2 and Figure 3. The structural model, along with standardized path coefficients and standard errors, is presented in Table 5.

#### 4.3.1. Non-Spectatorship Group (NSG)

In terms of the structural coefficients, the specific path coefficients for the NSG in the model revealed that cyberbullying victimization demonstrates statistically significant negative associations with key emotion management factors, supporting hypotheses *H2a*, *H2b*, *H2c*, and *H2d*. Specifically, cyberbullying victimization is found to significantly decrease other-emotion appraisal (β = −0.40, SE = 0.27, *p* < 0.001), indicating that students who have experienced cyberbullying victimization tend to exhibit reduced sensitivity and awareness of other people’s emotions. It also negatively impacts self-emotion appraisal (β = −0.33, SE = 0.24, *p* = 0.003), suggesting that these students may have difficulty recognizing and understanding their own emotions. Emotion expression (β = −0.22, SE = 0.19, *p* = 0.04) is likewise reduced, implying a decreased likelihood of openly expressing emotions and potential difficulties in emotional communication. Emotion control (β = −0.39, SE = 0.25, *p* = 0.001) is negatively influenced, indicating challenges in emotional control and regulation.

Next, in our investigation of the relationships between the four emotion management factors—other-emotion appraisal, self-emotion appraisal, emotion expression, and emotion control—and the four key outcomes—health-related quality of life, GPA, absences, and school-related offenses—we uncovered notable associations. Specifically, other-emotion appraisal displayed a negative association with health-related quality of life (β = −0.18, SE = 0.27, *p* = 0.04), indicating that students with heightened sensitivity to others’ emotions may prioritize the well-being of others over their own. Unexpectedly, it was positively associated with school-related offenses (β = 0.28, SE = 0.19, *p* = 0.009), suggesting that increased sensitivity to others’ emotions might lead to impulsive actions.

Self-emotion appraisal demonstrated a positive link with health-related quality of life (β = 0.24, SE = 0.24, *p* = 0.02), in line with our expectations. However, it also exhibited positive associations with absences (β = 0.41, SE = 0.25, *p* < 0.001) and school-related offenses (β = 0.19, SE = 0.18, *p* = 0.02), implying that students with heightened self-awareness of their emotions may encounter challenges related to attendance and impulsivity. Emotion expression showed positive associations with health-related quality of life (β = 0.21, SE = 0.13, *p* = 0.03) and school-related offenses (β = 0.32, SE = 0.32, *p* = 0.01) but unexpectedly displayed a negative association with GPA (β = −0.21, SE = 0.11, *p* = 0.01), suggesting that open emotional expression may divert academic focus. Emotion control exhibited a negative association with health-related quality of life (β = −0.27, SE = 0.16, *p* = 0.005), indicating that harmed emotion control can potentially impact life quality negatively. These results provided empirical evidence for *H3* and *H4*.

#### 4.3.2. Spectatorship-Adherent Group (SG)

In our analysis of the SG samples, we found several significant associations. First, cyberbullying victimization was positively associated with perceived self-emotion appraisal (β = 0.33, SE = 0.20, *p* = 0.004) and emotion control (β = 0.23, SE = 0.35, *p* = 0.02). However, there were no significant associations between cyberbullying victimization and other-emotion appraisal or emotion expression. With respect to the associations between emotion management factors and other consequential outcomes, in contrast to the NSG results, other-emotion appraisal showed a positive association with health-related quality of life (β = 0.20, SE = 0.21, *p* = 0.03) among SG students. Additionally, perceived self-emotion appraisal displayed unique associations. It was negatively associated with absences (β = −0.15, SE = 0.18, *p* = 0.04), implying that students with heightened self-awareness of their emotions, possibly nurtured through spectatorship experiences, have fewer school absences. Furthermore, perceived self-emotion appraisal exhibited positive links with both health-related quality of life (β = 0.39, SE = 0.25, *p* = 0.001) and GPA (β = 0.23, SE = 0.13, *p* = 0.01), indicating that those in tune with their emotions tend to have better well-being and academic performance.

Similarly to the other group, emotion expression was positively associated with school-related offenses (β = 0.22, SE = 0.17, *p* = 0.02). However, emotion expression also displayed a positive association with health-related quality of life (β = 0.31, SE = 0.26, *p* = 0.006), implying that open emotional expression may contribute to a higher quality of life. In contrast, emotion control was negatively associated with school-related offenses (β = −0.20, SE = 0.32, *p* = 0.02), suggesting that students with effective emotional control skills were less likely to be involved in school-related disciplinary incidents. Emotion control also exhibited positive associations with both health-related quality of life (β = 0.29, SE = 0.26, *p* = 0.002) and GPA (β = 0.25, SE = 0.33, *p* = 0.03), highlighting that effective emotional control can enhance overall well-being and academic performance. All other paths were found to be statistically non-significant.

The latent means of the SG samples were estimated by constraining both factor loadings and intercepts to be equal across groups, while setting the latent means of the NSG to zero ([30]). The results revealed that, within the emotion management factors, the latent means of other-emotion appraisal (0.43, *p* < 0.001), perceived self-emotion appraisal (0.27, *p* = 0.01), and emotion control (0.41, *p* < 0.001) were significantly higher in the SG compared to the latent means of NSG students. Additionally, the results demonstrated that, for the SG, the latent means of health-related quality of life (0.44, *p* < 0.001) and GPA (0.18, *p* = 0.02) were significantly higher while the latent mean of school-related offenses (−0.93, *p* < 0.001) was significantly lower in comparison to the latent means of NSG students. The latent means of other factors showed no significant differences between the two groups. These results collectively validated *H5*. 

## 5. Discussion

### 5.1. Theoretical Implications

#### 5.1.1. Cyberbullying and Impaired Emotion Management

This study presents substantial theoretical insights and contributions to the fields of emotion management (e.g., [15]; [64]) and youth health (e.g., [3]; [14]; [41]; [44]). It highlights a critical finding regarding the adverse effects of cyberbullying on emotional appraisal, impacting both self-perception and the perception of others. Victims grapple with the challenge of comprehending and connecting with the emotional experiences of their peers, a phenomenon potentially linked to desensitization resulting from the pervasive negativity prevalent in the online environment ([47]; [53]). Moreover, the study reveals the positive correlations between cyberbullying and occurrences of school-related absences and offenses, highlighting the difficulties faced by victims in evaluating their and others’ emotional states amid ongoing victimization. Interestingly, we also observe a positive impact on health-related quality of life, especially in association with self-emotion appraisal and emotion expression. However, it is important to explore the potential consequences of this improved quality of life as it may indeed be rooted in hubristic perceptions induced by cyberbullying ([52]).

In addition, our findings shed light on how cyberbullying inhibits emotion expression, obstructing adolescents from seeking support and engaging in healthy emotional communication. Over time, this inhibition contributes to an increase in school-related offenses. Interestingly, we also observe a positive impact on health-related quality of life, especially in association with self-emotion appraisal and emotion expression. However, it is crucial to explore the potential consequences of this improved quality of life, including the possibility of heightened overconfidence ([58]) and a sense of hubristic fulfillment through negative behavior ([6]), which could potentially lead to a decline in academic performance. The impaired health-related quality of life associated with compromised emotion control lends support to the argument that the elevated perceptions of life quality, linked to self-emotion appraisal and emotion expression, may indeed be rooted in hubristic perceptions induced by cyberbullying ([52]). Interestingly, the positive association between perceived self-emotion appraisal and absenteeism or disciplinary issues in the non-spectatorship group was initially unexpected. This might suggest that heightened self-focus, when unchecked, could lead to avoidance (e.g., absences) or impulsive reactions that translate into disciplinary incidents.

Consequently, our research emphasizes the heightened emotional reactivity and the concomitant challenges in emotion control and regulation instigated by cyberbullying among adolescents. This emphasizes the urgent need for targeted interventions, particularly for economically disadvantaged adolescents, to equip them with effective strategies for emotion regulation. In addition, our findings shed light on how cyberbullying inhibits perceived emotion expression among the economically disadvantaged adolescents in our sample, impeding their ability to seek support and engage in healthy emotional communication. Adolescents facing economic hardships often contend with added stressors ([36]; [44]; [55]) and may have reduced access to emotional support ([7]; [51]). These compounding factors exacerbate the emotional distress caused by online harassment. Implementing such interventions holds the promise of mitigating the detrimental effects of cyberbullying and enhancing the emotional resilience and overall well-being of adolescents.

#### 5.1.2. The Emotional Benefits of Sport Spectatorship

Our study explores the potential of spectator sports involvement as an intervention to counter the negative effects of cyberbullying on adolescents. Unlike prior research that emphasized the benefits of active sports participation (e.g., [2]; [24]; [33]; [50]; [63]), our work explores the uncharted territory of sports viewership as a cost-effective and meaningful intervention tool for economically disadvantaged adolescents. In the group of students with spectator sports involvement, we found intriguing associations between cyberbullying victimization, self-emotion appraisal, and emotion control. Cyberbullying victimization was positively connected with perceived self-emotion appraisal and perceived emotion control, indicating that those who engaged in spectator sports possessed perceived enhanced emotional coping mechanisms, allowing them to better understand and manage their emotions in the face of challenges such as cyberbullying. Spectatorship experiences expose students to a variety of emotional expressions, both from athletes and fellow spectators ([10]; [39]). This heightened perceived emotional self-awareness, and expression opportunities gained through these experiences may help reduce school absences, enhance overall well-being, and contribute to psychological health ([12]).

Emotion expression was positively associated with school-related offenses, possibly due to the emotional intensity experienced during sports events ([4]; [38]). However, it was also positively associated with health-related quality of life, suggesting that open emotional expression fosters a more supportive and socially engaging environment ([34]), ultimately leading to an improved quality of life ([49]). Emotion control had a negative association with school-related offenses, emphasizing the potential role of effective emotional control skills cultivated through sport spectatorship in reducing disciplinary incidents. This aligns with the theory of social learning ([50]; [57]), suggesting that students internalize positive behavioral norms observed in the sports environment. Emotion control also exhibited positive associations with health-related quality of life and GPA, underlining the advantages of effective emotional regulation in reducing emotional distress and improving academic success.

The findings emphasize the importance of addressing cyberbullying’s impact on perceived emotion management, aligning with the broader literature showing that perceived emotional competencies can significantly buffer negative stressors ([59]). Tripon’s work underscores the role of emotional intelligence in navigating high-pressure domains, and our results similarly indicate that fostering self-awareness and regulation can help adolescents navigate the complexities of cyberbullying. Additionally, the sense of belonging afforded by spectator sports can serve as a vital protective factor. Recent work highlights that adolescents with strong belonging in offline communities report fewer negative psychological outcomes, an effect likely amplified when facing online victimization ([17]).

Our findings emphasize the profound emotional advantages of passive engagement in spectator sports, which goes beyond general social engagement by offering a structured environment where adolescents regularly observe emotional highs and lows. Through wins, losses, and communal support, adolescents can repeatedly ‘practice’ identifying and regulating emotions in a vicarious manner, fostering improved emotion regulation skills. This distinctive aspect of spectatorship—compared to broad social interaction—may equip them to manage their emotions more effectively in the face of adversity such as cyberbullying ([1]). We highlight the valuable and uniquely cost-effective role of sports viewership in enhancing economically disadvantaged adolescents’ perceived emotional well-being. Because these youth often have fewer resources and may lack access to mental health services, the communal and emotionally rich environment of sports spectatorship can serve as a readily available outlet for social bonding and stress relief.

A noteworthy contrast emerges when comparing these findings to the non-spectatorship group (NSG). Students who did not regularly watch sports reported heightened difficulties in emotion control and a stronger link between cyberbullying and negative school-related outcomes. Unlike their counterparts in the spectatorship-adherent group, NSG students did not appear to benefit from the emotional release or communal support that sports viewership may provide. These differences underscore the potential protective role of even passive sports engagement—especially for economically disadvantaged adolescents who may have limited options for other emotion-focused outlets.

### 5.2. Practical Implications

The results provide important managerial implications. One key takeaway from our research is the importance of tailoring emotion management strategies to address the specific facets of emotion appraisal, expression, and control. Schools and mental health professionals can develop tailored programs targeting each of these factors separately, enabling adolescents to build a comprehensive set of emotional coping skills. Recognizing that emotion management is not a one-size-fits-all concept, educational institutions can design precise initiatives that cater to the diverse emotional needs of students, ultimately enhancing their resilience in the face of cyberbullying and creating a more supportive environment for emotional development.

Furthermore, our findings emphasize the emotional benefits of sport spectatorship, suggesting it can be a powerful and cost-effective intervention tool. Schools and community organizations may consider collaborating with local sports events, providing adolescents with opportunities to attend matches, games, or tournaments. Sports organizations could partner with schools to sponsor free or low-cost admission to local sporting events or create streaming partnerships that grant economically disadvantaged students easy access. Policymakers might also support grant programs that integrate structured viewing sessions into after-school or community-based programs. These experiences expose them to a broad spectrum of emotional expressions, both from athletes and fellow spectators, fostering emotional self-awareness and expression. Integrating such initiatives into school curricula not only contributes to students’ emotional well-being but also expands their horizons beyond the classroom, offering them a holistic educational experience ([28]). As such, this study lays the foundation for innovative strategies to prevent cyberbullying and enhance adolescent emotional well-being. By addressing the intricacies of emotion management and harnessing the potential of spectatorship sports, educators and stakeholders can craft more effective approaches to equip adolescents with the emotional tools they need to thrive in today’s digital world.

### 5.3. Limitations and Future Suggestions

There are several limitations that merit consideration for future investigations. First, the results are derived from a single school district. Despite the considerable size of the school and student population, the utilized samples may be overly homogenous, restricting the generalizability of the findings. Second, the definition of economic disadvantage could vary across states and cultures; therefore, incorporating more diverse samples would enhance the validity of the results. Our sample spans ages 12 to 19, a broad range that may encompass significant developmental differences in cognitive and emotional maturity. Although this enabled us to capture a diverse adolescent population, future research should differentiate early adolescents (12–14) from older adolescents (15–19) to identify age-specific patterns in emotion management and cyberbullying experiences. Additionally, the reliance on self-report measures can introduce social desirability or recall biases. Future investigations using multi-informant approaches (e.g., teacher ratings, peer assessments) or physiological data could provide more objective measures of emotional competencies. Our sample exclusively comprises economically disadvantaged adolescents from a single school district in Texas, which restricts the generalizability of our findings. The heightened impact of cyberbullying on emotional well-being observed here may not translate directly to adolescents with more financial resources.

Future studies should aim to enhance the internal validity of outcomes by further isolating the effects of sport viewership. This can be achieved by explicitly considering regular engagement, such as participation more than once per week over the past year, in both school-related and non-school-curriculum-related sports and physical activities. Adopting this approach would enable researchers to focus exclusively on the impact of sport viewership while effectively controlling for active sport participation and other forms of regular physical activity.

## Figures and Tables

**Figure 1 behavsci-15-00555-f001:**
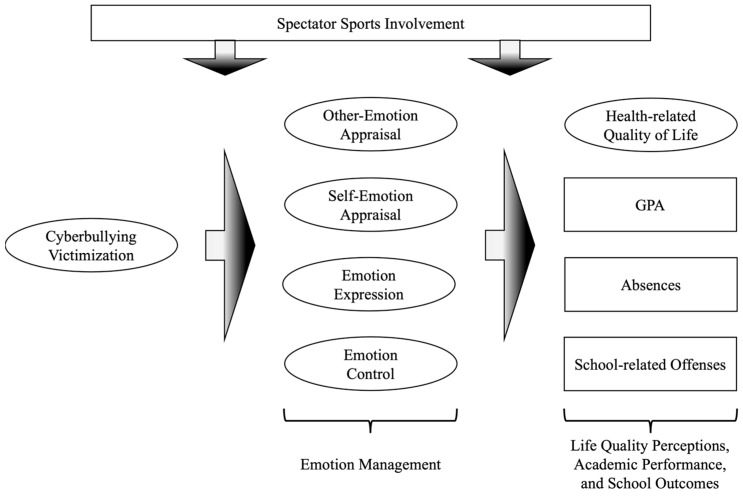
Hypothesized research model.

**Figure 2 behavsci-15-00555-f002:**
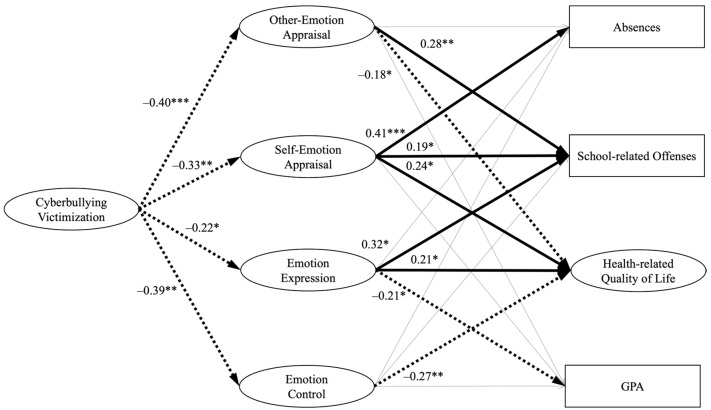
Summaries of structural coefficients for non-spectatorship group (NSG). Note: ** p* < 0.05; ** *p* < 0.01; *** *p* < 0.001. All four emotion management variables, including other-emotion appraisal, self-emotion appraisal, emotion expression, and emotion control, along with all four endogenous variables, which include absences, school-related offenses, health-related quality of life, and GPA, are allowed to freely covary, respectively ([30]). In the graphical representation, thick lines signify significant path coefficients, while thin lines indicate non-significant path coefficients. Dotted lines represent negative path coefficients.

**Figure 3 behavsci-15-00555-f003:**
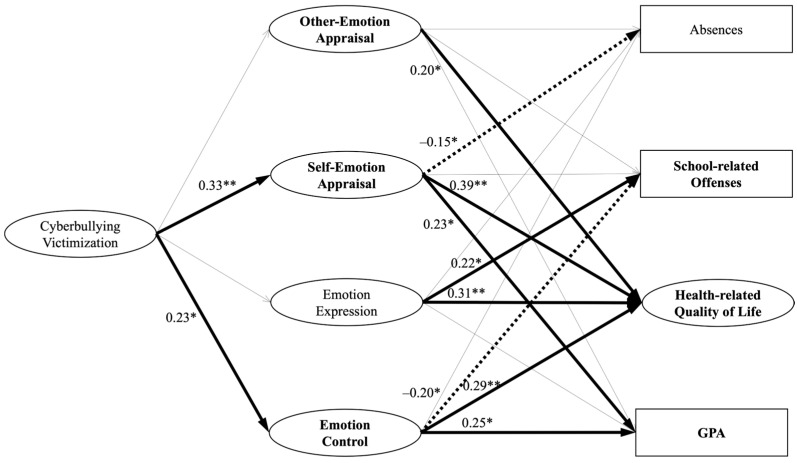
Summaries of structural coefficients for spectatorship-adherent group (SG). Note: ** p* < 0.05; ** *p* < 0.01. All four emotion management variables, including other-emotion appraisal, self-emotion appraisal, emotion expression, and emotion control, along with all four endogenous variables, which include absences, school-related offenses, health-related quality of life, and GPA, are allowed to freely covary, respectively ([30]). In the graphical representation, thick lines signify significant path coefficients, while thin lines indicate non-significant path coefficients. Dotted lines represent negative path coefficients. Factor names in bold font highlight significant mean differences between the two groups.

**Table 1 behavsci-15-00555-t001:** Means (M), standard errors (SE), and factor loadings (λ).

Variables	Items	*M*	*SE*	*λ*
Cyberbullying Victimization	*I have been the target of online bullying or mean messages from other kids.*	3.19	0.44	0.64
	*I have received threatening messages or calls from other kids on my phone.*	2.81	0.39	0.58
	*Other kids have posted hurtful or mean things about me online.*	2.72	0.44	0.65
Other-emotion Appraisal	*I can often tell how others are feeling by their expressions and behavior.*	4.62	0.09	0.74
	*I’m good at understanding how others feel.*	4.75	0.09	0.73
	*I can usually tell when someone needs help with their feelings.*	4.46	0.08	0.77
Self-emotion Appraisal	*I often think about why I feel the way I do.*	4.42	0.11	0.69
	*It’s easy for me to understand how I feel.*	4.37	0.14	0.67
	*I pay a lot of attention to my feelings.*	4.58	0.10	0.69
Emotion Expression	*I feel comfortable talking about my feelings with friends and family.*	3.25	0.10	0.71
	*Sharing my emotions with others is something I do easily.*	3.53	0.11	0.72
	*I’m open to discussing how I feel with those I trust.*	3.66	0.10	0.71
Emotion Control	*I try to control my thoughts and not worry too much about things.*	3.73	0.13	0.68
	*I can often keep my emotions in check, even in stressful situations.*	3.13	0.12	0.70
	*I find it easy to control my feelings.*	3.35	0.12	0.70
Health-related Quality of Life	*Felt sad (reversed).*	2.57	0.47	0.54
	*Felt lonely (reversed).*	2.33	0.44	0.58
	*Felt fit and well.*	4.59	0.11	0.74
	*Felt full of energy.*	4.32	0.11	0.75
	*Had enough time for yourself.*	5.74	0.09	0.83
	*Been able to do the things that you want in your free time.*	5.98	0.14	0.66
	*Parents/guardians treated you fairly.*	4.93	0.13	0.68
	*Had fun with your friends.*	5.16	0.17	0.61
	*Felt confident at school.*	4.27	0.11	0.71
	*Been able to pay attention at school.*	4.51	0.14	0.70

**Table 2 behavsci-15-00555-t002:** Correlations matrix.

	1	2	3	4	5	6	7	8	9
Cyber. Victimization	–								
Other-emotion.	−0.41/0.11	–							
Self-emotion.	−0.44/0.42	0.76/0.51	–						
Emotion Expression	−0.31/0.18	0.60/0.49	0.61/0.55	–					
Emotion Control	−0.39/0.31	0.58/0.51	0.73/0.38	0.57/0.48	–				
Absences	0.37/0.04	0.02/0.14	0.48/−0.11	0.18/0.17	0.10/0.17	–			
Offenses	0.41/0.11	0.37/0.21	0.15/0.04	0.26/0.19	0.11/−0.05	0.03/−0.02	–		
Life Quality	−0.10/0.09	−0.21/0.03	0.32/0.27	0.34/0.28	−0.24/0.21	0.01/0.07	−0.17/0.14	–	
GPA	−0.21/−0.01	0.14/0.01	0.21/0.34	−0.23/0.07	0.02/0.29	−0.02/−0.01	−0.23/−0.01	0.04/0.17	–

Note: Non-spectatorship group (NSG)/spectatorship-adherent group (SG).

**Table 3 behavsci-15-00555-t003:** Summary of fit statistics for testing measurement invariance.

Model	χ^2^	*df*	RMSEA	CFI	Model Comparison	Difference in χ^2^/*df*	Difference in CFI	*p* Value
1. Configural invariance (unconstrained)	985.44	260	0.06	0.91	–	–	–	–
2. Factor loadings invariant	1007.21	272	0.07	0.90	2 vs. 1	21.77/12	0.01	0.13
3. Factor loadings and intercepts invariant	1043.82	284	0.07	0.90	3 vs. 1	58.38/24	0.01	<0.001

Note. RMSEA = root mean squared error of approximation; CFI = comparative fit index.

**Table 4 behavsci-15-00555-t004:** Summary of fit statistics for testing structural invariance.

Model	Model Fit Indices	Model Comparison in χ^2^/*df*	Structural Coefficients	
			NSG	SG
Unconstrained	χ^2^ (642) = 2381.8; C = 0.92, R = 0.06, S = 0.07	–	–	–
Structural weight	χ^2^ (662) = 2464.2; C = 0.90, R = 0.07, S = 0.08	82.37/20	–	–
Cyber. Victim. → Other-emotion	χ^2^ (643) = 2384.9; C = 0.91, R = 0.07, S = 0.07	3.12/1	−0.40 ***	0.06
Cyber. Victim. → Self-emotion	χ^2^ (643) = 2384.7; C = 0.91, R = 0.07, S = 0.07	2.85/1	−0.33 **	0.33 **
Cyber. Victim. → Expression	χ^2^ (643) = 2385.9; C = 0.91, R = 0.07, S = 0.07	4.03/1	−0.22 *	0.15
Cyber. Victim. → Control	χ^2^ (643) = 2384.7; C = 0.91, R = 0.07, S = 0.07	2.58/1	−0.39 **	0.23 *
Other-emotion → Absences	χ^2^ (643) = 2381.5; C = 0.92, R = 0.06, S = 0.07	0.03/1	0.01	0.02
Other-emotion → Offenses	χ^2^ (643) = 2381.1; C = 0.92, R = 0.06, S = 0.07	0.63/1	0.28 **	0.16
Other-emotion → Life Quality	χ^2^ (643) = 2385.3; C = 0.91, R = 0.07, S = 0.07	3.44/1	−0.18 *	0.20 *
Other-emotion → GPA	χ^2^ (643) = 2382.1; C = 0.92, R = 0.06, S = 0.07	0.31/1	0.10	0.01
Self-emotion → Absences	χ^2^ (643) = 2383.8; C = 0.91, R = 0.07, S = 0.07	2.01/1	0.41 ***	−0.15 *
Self-emotion → Offenses	χ^2^ (643) = 2381.1; C = 0.92, R = 0.06, S = 0.07	0.74/1	0.19 *	0.05
Self-emotion → Life Quality	χ^2^ (643) = 2382.1; C = 0.92, R = 0.06, S = 0.07	0.23/1	0.24 *	0.39 **
Self-emotion → GPA	χ^2^ (643) = 2382.2; C = 0.92, R = 0.06, S = 0.07	0.32/1	0.10	0.23 *
Expression → Absences	χ^2^ (643) = 2382.1; C = 0.92, R = 0.06, S = 0.07	0.39/1	0.01	0.12
Expression → Offenses	χ^2^ (643) = 2382.0; C = 0.92, R = 0.06, S = 0.07	0.35/1	0.32 *	0.22 *
Expression → Life Quality	χ^2^ (643) = 2382.0; C = 0.92, R = 0.06, S = 0.07	0.31/1	0.21 *	0.31 **
Expression → GPA	χ^2^ (643) = 2384.6; C = 0.91, R = 0.07, S = 0.07	2.70/1	−0.21 *	0.04
Control → Absences	χ^2^ (643) = 2381.1; C = 0.92, R = 0.06, S = 0.07	0.07/1	0.10	0.13
Control → Offenses	χ^2^ (643) = 2385.6; C = 0.91, R = 0.07, S = 0.07	3.71/1	0.12	−0.20 *
Control → Life Quality	χ^2^ (643) = 2384.3; C = 0.91, R = 0.07, S = 0.07	2.46/1	−0.27 **	0.29 **
Control → GPA	χ^2^ (643) = 2380.5; C = 0.92, R = 0.06, S = 0.07	1.22/1	0.02	0.25 *

Note: * *p* < 0.05. ** *p* < 0.01. *** *p* < 0.001. C = CFI; R = RMSEA; S = SRMR.

**Table 5 behavsci-15-00555-t005:** The structural model with standardized path coefficients (β) and standard errors (SE).

	Moderating Effects	Non-Spectatorship Group (NSG)			Spectatorship-Adherent Group (SG)		
Relationship		*β*	*SE*	*p*-Value	*β*	*SE*	*p*-Value
*From Cyber. Victimization to Emotion Management*							
Cyber. Victimization → Other-emotion Appraisal		−0.40 ***	0.27	<0.001	0.06	0.30	0.48
Cyber. Victimization → Self-emotion Appraisal		−0.33 **	0.24	0.003	0.33 **	0.20	0.004
Cyber. Victimization → Emotion Expression		−0.22 *	0.19	0.04	0.15	0.42	0.14
Cyber. Victimization → Emotion Control		−0.39 **	0.25	0.001	0.23 *	0.35	0.02
*From Other-emotion Appraisal to DVs*							
Other-emotion Appraisal → Absences		0.01	0.47	0.94	0.02	0.24	0.68
Other-emotion Appraisal → School-related Offenses		0.28 **	0.19	0.009	0.16	0.37	0.09
Other-emotion Appraisal → Health-related Quality of Life		−0.18 *	0.27	0.04	0.20 *	0.21	0.03
Other-emotion Appraisal → GPA		0.10	0.32	0.71	0.01	0.39	0.67
*From Self-emotion Appraisal to DVs*							
Self-emotion Appraisal → Absences		0.41 ***	0.25	<0.001	−0.15 *	0.18	0.04
Self-emotion Appraisal → School-related Offenses		0.19 *	0.18	0.02	0.05	0.27	0.59
Self-emotion Appraisal → Health-related Quality of Life		0.24 *	0.24	0.02	0.39 **	0.25	0.001
Self-emotion Appraisal → GPA		0.10	0.25	0.32	0.23 *	0.13	0.01
*From Emotion Expression to DVs*							
Emotion Expression → Absences		0.01	0.45	0.92	0.12	0.43	0.29
Emotion Expression → School-related Offenses		0.32 *	0.32	0.01	0.22 *	0.17	0.02
Emotion Expression → Health-related Quality of Life		0.21 *	0.13	0.03	0.31 **	0.26	0.006
Emotion Expression → GPA		−0.21 *	0.11	0.01	0.04	0.32	0.57
*From Emotion Control to DVs*							
Emotion Control → Absences		0.10	0.27	0.52	0.13	0.41	0.25
Emotion Control → School-related Offenses		0.12	0.35	0.37	−0.20 *	0.32	0.02
Emotion Control → Health-related Quality of Life		−0.27 **	0.16	0.005	0.29 **	0.26	0.002
Emotion Control → GPA		0.02	0.38	0.82	0.25 *	0.33	0.03

Note: * *p* < 0.05; ** *p* < 0.01; *** *p* < 0.001.

## Data Availability

The data presented in this study are available on request from the corresponding author. The data are not publicly available due to privacy and ethical restrictions, as they include health-related information from human subjects. All personally identifiable information has been permanently removed.

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
