# Peer review of "Cracking the Code of Cyberbullying Effects: The Spectator Sports Solution for Emotion Management and Well-Being Among Economically Disadvantaged Adolescents"

_behavsci, 2025, doi:10.3390/bs15040555_

Round 1
Reviewer 1 Report
Comments and Suggestions for Authors
This paper explores a timely and relevant issue—how sports spectatorship may serve as a coping mechanism for economically disadvantaged adolescents facing cyberbullying. The study is well-structured, and the theoretical framework is grounded in emotional regulation and social support theories. However, some areas require clarification and refinement to strengthen the manuscript’s contribution to the literature.
- The paper should further elaborate on how sports spectatorship specifically contributes to emotional regulation beyond general social engagement.
- More discussion on how economically disadvantaged adolescents may uniquely benefit from sports engagement is needed.
- Details about sample selection and recruitment should be more transparent—was there an effort to ensure diversity in the sample
- If survey-based, more information is needed on the reliability and validity of measures. If it combined with qualitative, consider adding more direct quotations or thematic analysis to reinforce findings.
- While the findings are promising, the discussion should explicitly link results back to prior research. Some references: Tripon, C. (2023). Navigating the STEM jungle of professionals: unlocking critical competencies through emotional intelligence. J Educ Sci Psychol, 13, 34-47.
- Were there any surprising or contradictory findings? A critical reflection on these would enhance the depth of analysis.
- The paper could expand on actionable recommendations for policymakers, educators, and sports organizations.
Reviewer 2 Report
Comments and Suggestions for Authors
"Emotion Management in the Face of Cyberbullying: The Role of Sport Spectatorship Among Economically Disadvantaged Adolescents" is an original and well-structured study that follows the journal's organizational guidelines. It is well-written, relevant, and brings valuable contributions to the field. The topic is of broad interest, given the increasing awareness of mental health among adolescents and the impact of cyberbullying. The originality of the study lies in the chosen population and its connection to sport spectatorship.
There are several positive aspects of the article under review. The theoretical framework supports the proposed hypotheses with updated and relevant literature. The research question is well-defined, and the study makes a valuable contribution to the literature by exploring the nuanced roles of perceived emotional competencies, cyberbullying, socioeconomic factors, and adolescents' perceived mental health.). One of the study’s strengths is the use of advanced statistical techniques, such as structural equation modeling, ensuring methodological rigor. The sample size is adequate, enhancing statistical power. The procedures are well-described, making the study replicable and strengthening its validity.
However, some limitations should be noted. The study is restricted to a single school, as acknowledged by the authors, which compromises the generalizability of the findings. Additionally, the use of self-report measures may introduce biases in data interpretation, and the authors should explicitly mention this as a limitation.
The results are presented in a comprehensive and impartial manner, further enhancing the study’s credibility. The findings offer valuable insights into adolescents' perceived mental health and protective factors. However, the discussion could be refined to better clarify that the study evaluates perceived emotional competencies rather than actual competencies, as well as adolescents' perceived mental health. For example, the statement:"Cyberbullying victimization was positively connected with self-emotion appraisal and emotion control, indicating that those engaged in spectator sports possessed enhanced emotional coping mechanisms, allowing them to better understand and manage their emotions in the face of challenges, such as cyberbullying."
should be revised to:
"Cyberbullying victimization was positively connected with perceived self-emotion appraisal and perceived emotion control, suggesting that those engaged in spectator sports perceive themselves as having stronger emotional coping mechanisms, which leads them to believe they can understand and manage their emotions in the face of challenges such as cyberbullying."
Furthermore, since the studied population consists exclusively of economically disadvantaged adolescents, the conclusions should not be generalized to all adolescents. Statements such as:
"In addition, our findings shed light on how cyberbullying inhibits emotion expression, obstructing adolescents from seeking support and engaging in healthy emotional communication. Over time, this inhibition contributes to an increase in school-related offenses. Interestingly, we also observe a positive impact on health-related quality of life, especially in association with self-emotion appraisal and emotion expression."
should be refined to reflect these limitations.
Reviewer 3 Report
Comments and Suggestions for Authors
This study investigates the relationship between cyberbullying victimization among economically disadvantaged students, their emotion management, well-being indicators, and the role of sport spectatorship. The literature review provides a comprehensive overview of the studied phenomena, forming the basis for the hypotheses. The study sample is thoroughly described (N=846). The methods are appropriate for the research objectives. The results are presented clearly and comprehensively. In the discussion, the authors justify their findings and compare them with existing research. The conclusion offers practical recommendations, acknowledges limitations, and outlines future research directions. The novelty and relevance of exploring the role of sport spectatorship in coping with the consequences of cyberbullying victimization are particularly noteworthy.
The study is well-executed and meets the fundamental requirements for scientific research. However, several points require clarification.
- The terminology used, specifically "emotion management," raises questions that are not addressed in the theoretical framework. How does this term relate to emotional intelligence?
- The emotion management scale, whose factor structure the authors examine in their first hypothesis, is based on a measure of emotional intelligence. How does this scale differ from the underlying emotional intelligence measure? What is the need for a new measurement instrument if it does not differ significantly? The proposed four factors of emotion management appear to represent a classical approach.
- Аlso I suggest thinking about the title of the article to convey the core meaning related to emotion management and well-being.
Reviewer 4 Report
Comments and Suggestions for Authors
I was very curious to read this study, and I think it has a good contribution to the development of the current literature. Here are some comments that I feel like sharing with you:
- The 'abtract should report information on nationality, sociodemographic characteristics of the sample, and type of study/research design. This will facilitate reference by other authors.
- Since we are dealing with adolescents, I ask you to elaborate on the topic of the need for belonging as a protective factor in the psychological and relational development of minors, balancing the online and real-world context ( 10.1007/s10578-023-01516-x).
- I don't quite understand how being a spectacular can buffer the effects of cyber-bullying. Also, the literature on the relationship between CB and emotional management should be expanded and updated. I think a recent contribution by Rosalba Morese on cyber-bullying, empathy and distress might be of interest to you.
- Sport spectator is not very clear whether you mean a spectator at the stadium or via TV and social or both. For me it is interesting if this aspect is clarified from the introduction.
Round 2
Reviewer 1 Report
Comments and Suggestions for Authors
very good article